

# Spatial and temporal dynamics of the bacterial community under experimental warming in field-grown wheat

Jing Wang[1,2], Shuaimin Chen[1], Ruibo Sun[1], Binbin Liu[1,3], Tatoba Waghmode[1] and Chunsheng Hu[1,3]

[1] Key Laboratory of Agricultural Water Resources, Hebei Laboratory of Agricultural Water-Saving, Center for Agricultural Resources Research, Institute of Genetic and Developmental Biology, The Chinese Academy of Sciences, Shijiazhuang, Hebei, China
[2] University of Chinese Academy of Sciences, Beijing, China
[3] Xiong'an Institute of Innovation, Chinese Academy of Sciences, Xiong'an New Area, China

## ABSTRACT

Climate change may lead to adverse effects on agricultural crops, plant microbiomes have the potential to help hosts counteract these effects. While plant–microbe interactions are known to be sensitive to temperature, how warming affects the community composition and functioning of plant microbiomes in most agricultural crops is still unclear. Here, we utilized a 10-year field experiment to investigate the effects of warming on root zone carbon availability, microbial activity and community composition at spatial (root, rhizosphere and bulk soil) and temporal (tillering, jointing and ripening stages of plants) scales in field-grown wheat (*Triticum aestivum* L.). The dissolved organic carbon and microbial activity in the rhizosphere were increased by soil warming and varied considerably across wheat growth stages. Warming exerted stronger effects on the microbial community composition in the root and rhizosphere samples than in the bulk soil. Microbial community composition, particularly the phyla Actinobacteria and Firmicutes, shifted considerably in response to warming. Interestingly, the abundance of a number of known copiotrophic taxa, such as *Pseudomonas* and *Bacillus,* and genera in *Actinomycetales* increased in the roots and rhizosphere under warming and the increase in these taxa implies that they may play a role in increasing the resilience of plants to warming. Taken together, we demonstrated that soil warming along with root proximity and plant growth status drives changes in the microbial community composition and function in the wheat root zone.

## INTRODUCTION

Earth's mean temperature has increased steadily over the past century and is predicted to further increase by 1.5 °C within the next two decades (*IPCC, 2021*). Most ecosystem models have suggested that warming can stimulate microbial decomposition of soil carbon and therefore produce positive feedback (*Allison, Wallenstein & Bradford, 2010*). Based on a long-term field warming experiment, it was extrapolated that continued warming will

Corresponding authors
Tatoba Waghmode,
tatobawaghmode@yahoo.com
Chunsheng Hu, cshu@sjziam.ac.cn

cause a loss of 190 petagrams of carbon by the end of the century, which is equivalent to the amount produced over the past two decades from fossil fuel emissions (*Melillo et al., 2017*). However, it is still a challenge to predict the contribution of soil to greenhouse gases under future climate scenarios due to unknown changes in soil nutrient pools and differences in microbial responses between soil locations (*Jansson & Hofmockel, 2020*).

Climate change could affect agricultural crops in a variety of ways. Elevated temperatures may cause severe cellular injury and cell death and lead to a decrease in plant growth and crop yield (*Abd El-Daim, Bejai & Meijer, 2014*). To cope with heat stress, plants make physiological adaptations by altering the expression of genes and the synthesis of proteins including heat shock proteins (HSPs) (*Wang et al., 2004*) and reactive oxygen species (ROS) (*Mittler, Finka & Goloubinoff, 2012*). Recent studies have revealed that plant-associated microorganisms play crucial roles in the performance of the host and are perceived as the plant's second genome (*Berg et al., 2014*). Plants may recruit plant-growth-promoting rhizobacteria (PGPR) in their root zone to promote growth or improve tolerance toward abiotic stress (*Chen et al., 2019*; *Yang, Kloepper & Ryu, 2009*). Although the effects of warming on the complexity of the network and keystone species of the microbial community in agricultural soil have been revealed recently (*Tian et al., 2022*), very little information is available concerning the adjustment of plant–microbe interactions to improve the resilience of crops to heat stress.

Prior studies have demonstrated that warming can directly affect microbial activity and composition by influencing processes such as respiration and the functioning of genes related to carbon and nitrogen cycling (*Chen et al., 2018*; *Roy Chowdhury et al., 2021*; *Söllinger et al., 2022*; *Waghmode et al., 2018*; *Xue et al., 2016*). The indirect effects of climate warming on the microbiome include changes in soil properties and nutrient cycling *via* root carbon inputs to the rhizosphere (*Wan et al., 2005*; *Zhang et al., 2016*). Warming has been identified as essential in affecting nutrient transformation processes in the surrounding soil by changing the quality and stoichiometry of root exudation in forest ecosystems (*Qiao et al., 2014*; *Wang et al., 2021*; *Yin et al., 2013*; *Zhang et al., 2016*). However, most of these investigations were conducted on forest soils; studies on the effects of warming on the microbial composition and activity in agricultural crop ecosystems are scarce.

Wheat (*T. aestivum*) is one of the most widely grown food crops worldwide (*Cianferoni, 2016*). The North China Plain (NCP, located at 114−121°E and 32−40°N) is one of the most important winter wheat-producing regions in China (*Wu et al., 2006*) and supplies more than 50% of China's total wheat production (*Qin et al., 2015*). Winter wheat in this region is normally planted in October and harvested at the end of May or early June of the next year. The stages of tillering, jointing and ripening are often taken as the important growth stages to investigate the effects of warming on plant growth (*Du et al., 2022*; *Yu et al., 2018*). In this study, we utilized a 10-year (from 2008 to 2018) warming experiment running on the North China Plain that consisted of control (ambient temperature) and warming treatments over the wheat growing season to determine the effects of experimental warming on soil dissolved organic carbon, soil microbial activity and community composition in the root, rhizosphere and bulk soils spanning various developmental stages (tillering, jointing and ripening) of field-grown wheat. We hypothesized that warming would (a) alter the

carbon availability and (b) microbial community composition in the root zone, and (c) these effects would be strongly affected by the proximity to the plant root and plant growth stage.

## MATERIAL AND METHODS

### Experimental site and design

The soil warming experiment was established at the Luancheng Agro-Ecosystem Experimental Station of the Chinese Academy of Sciences on the North China Plain, Hebei, China (37°53′N, 114°41′E) in 2008. The field was cultivated with local winter wheat (*Triticum aestivum* L.) cultivar 'Shixin 828'. The soil was classified as a sandy loam with a pH of 8.1, 15.1 g kg$^{-1}$ organic matter, and 1.1 g kg$^{-1}$ total N in the 0–20 cm soil layer (*Liu et al., 2016*). The control and warming treatments were set up in a randomized block design, each replicated three times with an individual plot size of 4 m × 4 m. The warmed plots were heated with three infrared heaters (1000 W, size of 2 m × 0.02 m) that were installed at the center of the plot 2 m above the ground, which were distributed equally in the 2 m long area, and the radiation area was 2 m × 2 m. In the control plots, "dummy" heaters were installed with no power to imitate shading effects (Fig. S1). The daily average temperature of the topsoil in the warming plots was 1.5 °C higher than that of the control plots. The nitrogen fertilizer was urea, half of which was applied before sowing in October and the other half in April of the following year. All phosphate fertilizers were applied at 65 P$_2$O$_5$ kg hm$^{-2}$.

### Plant and soil sampling

Plant root-zone (root, rhizosphere and bulk soil) samples were collected in November (Feekes growth stage 2–3, tillering stage), March(Feekes stage 6–7, jointing stage) and May (Feekes stage 11, ripening stage) during the wheat growing season. At each growth stage, root samples and rhizosphere soil of the control and warming treatments were randomly taken with three replicates using the method described previously (*Chen et al., 2019*). After gently shaking the roots to remove loosely bound soil clumps, the rhizosphere soil was carefully brushed out of the roots (*Clemensson-Lindell & Persson, 1992*). The roots were washed with sterile distilled water and used for endosphere bacterial community analyses. The sample collection method did not discriminate between microbial communities at the root surface and in the endosphere; therefore, we considered the root fraction as the 'root microbiome' (*Hu et al., 2018*). To sample bulk soil, three soil cores (0–20 cm soil depth) were randomly taken from each plot to form a composite sample. In total, three replicate plots from the warming treatment and three from the control were sampled at each growth stage. Roots, rhizosphere and bulk soil samples were stored at −80 °C before DNA extraction and at 4 °C for measurement of dissolved organic carbon (DOC) and enzyme activities.

### Soil properties and dehydrogenase activities

DOC is defined as the dissolved part of organic carbon in the soil. DOC in the rhizosphere was extracted with sterile distilled water and measured by an elemental analyzer (vario TOC;

Elementar Analysensysteme GmbH, Langenselbold, Germany). Soil pH was measured using a pH electrode (1:5, soil:water) (*Rayment & Higginson, 1992*), soil temperature and soil moisture were monitored continuously using T-type thermal couples and time domain reflectometry (TDR 100 system, USA), respectively, and data were recorded every hour using a data logger (CR10X, Campbell, USA) in control and warmed treatment plots.

The dehydrogenase enzyme activity is considered a marker of microbial activities in the soil and was assessed for the rhizosphere soils in the current study. Triplicate samples collected at the tillering, jointing and ripening stages from both the control and warming treatments were used for enzyme activity determination. The soil was passed through a two mm sieve and stored at 4 °C, and activity was measured within one week of sampling. In brief, 6 g of fresh soil and 60 mg of $CaCO_3$ were incubated with 1 ml of 3% triphenyl tetrazolium chloride (TTC) and 2.5 ml of deionized water at 37 °C for 24 h. Triphenylformazan (TPF), a product from the reduction of TTC, was extracted with methanol in a 100 ml volumetric flask, and the color intensity was measured at a wavelength of 485 nm (*Tabatabai, 1994*).

## DNA extraction, PCR amplification and sequencing

The DNA was extracted from 0.5 g of fresh root powder that was obtained by grinding with liquid nitrogen. Total DNA of the rhizosphere and bulk soil was extracted using an E.Z.N.A. Soil DNA Kit (Omega Biotek, Inc., Norcross, GA, USA) following the manufacturer's instructions. The concentration and quality of extracted DNA were determined using a NanoDrop spectrophotometer (NanoDrop-2000c Technologies, Inc., Wilmington, DE, USA), and extracted DNA was stored at −20 °C until further use.

The bacterial 16S rRNA gene (V3–V4 region; approximately 460 bp) was amplified with primers 341F (5′-CCTACGGGNGGCWGCAG-3′) and 785R (5′-GACTACHVGGGTATCTAATCC-3′) (*Yasir et al., 2015*). Overhanging bases were added to the primers to connect the Illumina sequencing adapters and dual-index barcodes in a second round of PCR. Each PCR was performed in a 25 µl mixture containing 12.5 µl of PCR Premix Ex Taq™ (Takara Biotech, Dalian, China), 1 µl of each primer (10 µM), and 1 µl of DNA template (approximately 20 ng DNA). The PCR conditions were as follows: 95 °C for 3 min; 25 cycles of 30 s at 95 °C, 30 s at 55 °C and 30 s at 72 °C, with a final extension at 72 °C for 10 min. The PCR products were visually examined on agarose gels and then purified with AMPure XP beads (Beckman Coulter, Inc., Brea, CA, USA) following the manufacturer's protocol. Subsequently, eight-cycle PCR was carried out to add Illumina sequencing adapters and dual-index barcodes to each sample, and then the PCR product was purified using AMPure beads. The libraries were then normalized according to the Nextera XT (Illumina) protocol, and samples were sequenced on a MiSeq PE300 platform (GENEWIZ, Suzhou, China).

## Sequence processing and analysis

The raw sequences were processed mainly with QIIME2 (2020.11) (*Bolyen et al., 2019*). The 16S rRNA gene sequences were quality filtered and denoised using DADA2, followed by the creation of amplicon sequence variants (ASVs) using the Deblur tool (*Callahan et al., 2016*).

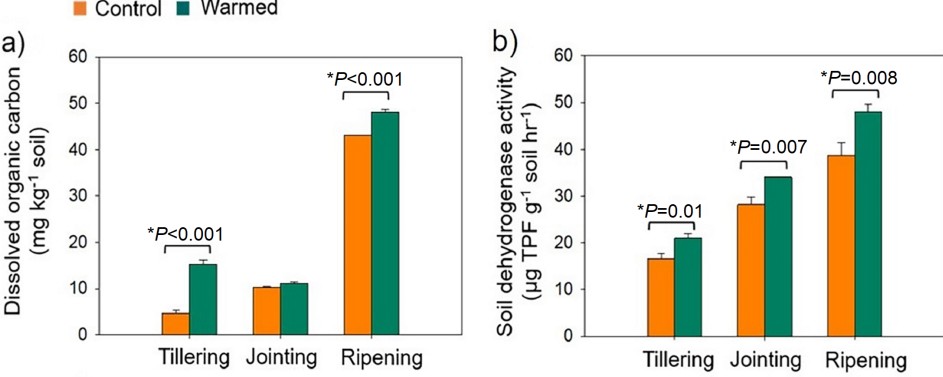

**Figure 1  Rhizosphere soil dissolved organic carbon (mg kg⁻¹ soil) (A) and soil dehydrogenase activity (µg TPF g⁻¹ soil⁻¹ hr⁻¹) (B) from the control and warmed treatments at three different wheat developmental stages.** An asterisk (*) indicates a significant difference at $P < 0.05$ (Student's $t$-test, 2 tailed). Error bars indicate values of mean ($n = 3$) and standard errors. TPF, triphenylformazan.

The taxonomic identities of the ASVs were obtained using the QIIME2 feature-classifier plugin (sklearn method) against the SILVA v.138 database (*Quast et al., 2013*). The alpha and beta diversity of the microbial communities were calculated within QIIME2 based on the standardized ASV table. PICRUSt2 (https://github.com/picrust/picrust2) was utilized to predict the functional potential of the microbial community (*Douglas et al., 2020*). Sequencing data were deposited into the European Nucleotide Archive under accession number PRJEB37653.

## Statistical analyses

All statistical analyses were carried out with SPSS 20.0 (IBM, Chicago, USA) and R v4.0.3 (*Team RC, 2014*). Student's $t$ test was used to compare the means of the control treatment and warming treatment at the $P < 0.05$ level. The R packages "ggplot2" and "pheatmap" were used to draw the point plots, bar plots, and heatmaps of bacterial diversity and community composition. Principal component analysis (PCA) was performed on the bacterial community at the genus level using the packages "vegan" and the results were visualized with "ggplot2" to determine the effects of warming and wheat developmental stage on the community structure in the root, rhizosphere and bulk soil.

## RESULTS

### Soil characteristics and microbial activity

Warming was simulated in the wheat field plots by using infrared heaters. The soil moisture was significantly lower in the warmed plots than in the control plots (Fig. S2). The DOC increased under warming treatments (Fig. 1A). DOC increased with wheat development and was found to be higher at the ripening stage. Dehydrogenase activity (*i.e.,* microbial activity) was higher ($P < 0.05$) under warming at all developmental stages (Fig. 1B).

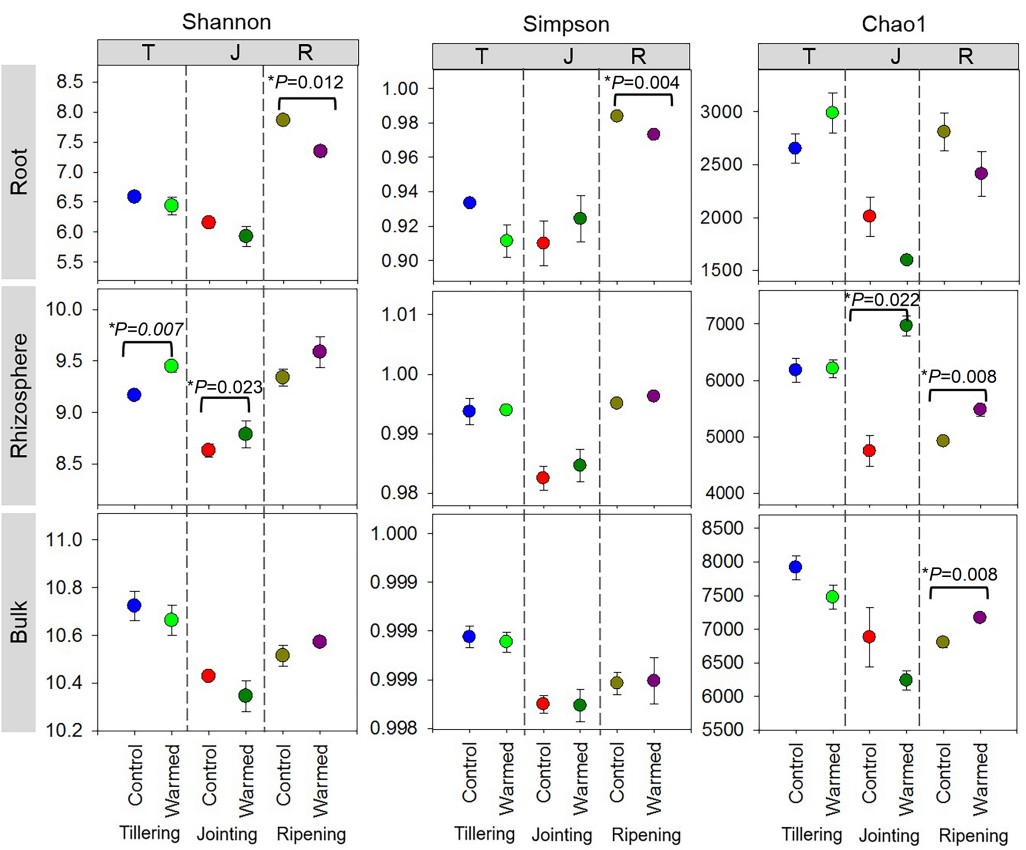

**Figure 2  Alpha diversity assessed by diversity (Shannon, Simpson) and richness (Chao1) in root, rhizosphere and bulk soil (based on ASV of the 16S rRNA gene) from the control and warmed treatments at three different wheat developmental stages.** An asterisk (*) indicates significance at $P < 0.05$ (Student's $t$-test, 2 tailed). Error bars indicate means ($n = 3$) and standard errors.

## Bacterial diversity and richness

High-throughput sequencing was carried out on the root, rhizosphere and bulk soil samples collected at the tillering, jointing and ripening stages of wheat grown under control and warming conditions. The effects of warming, root proximity and plant growth stages on bacterial diversity (Shannon and Simpson) and richness (Chao1) indexes were determined (Fig. 2). In the root compartment, bacterial diversity(Shannon index) generally decreased in the warmed plots compared to the control plots, and the Shannon and Simpson indexes were significantly higher at the ripening stage than at the tillering and jointing stages. In contrast, bacterial diversity and richness increased with warming, and the Shannon and Chao1 indexes were generally higher at the tillering and ripening stages in the rhizosphere soil. Alpha diversity also decreased in the bulk soil at the early wheat developmental stage but was not statistically significant.

Soil warming along with wheat development stage considerably influenced the root bacterial community when compared to the rhizosphere and bulk soil (Fig. 3, Fig. S3). The PCA showed greater separation for communities between the control and warmed

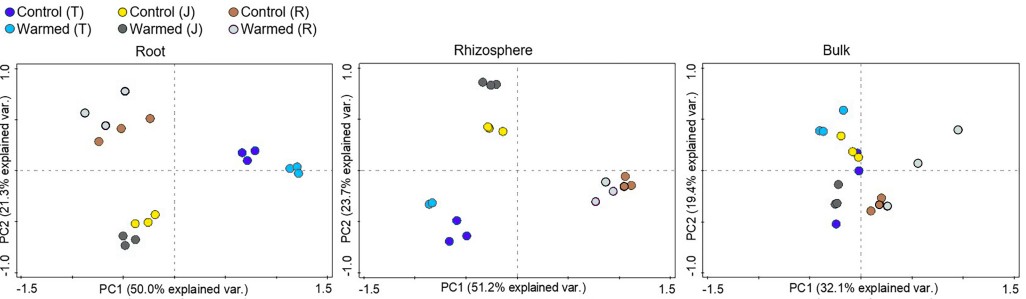

**Figure 3 Principal component analysis (PCA) of the genus microbial communities in root, rhizosphere and bulk soil from the control and warmed treatments at three different wheat developmental stages.** T, tillering; J, jointing; R, ripening.

plots in the root and rhizosphere compared with the bulk soil. In roots, PC1 and PC2 explained 50.0% and 21.3% of the variability in the bacterial community of samples from all wheat development stages. The separation of the bacterial community between the control and warmed plots decreased with the wheat development stage in root samples. In the rhizosphere, PC1 and PC2 accounted for 51.2% and 23.7% of the variance in the data, respectively. In bulk soil, there was no significant separation between treatments and the development stage.

## Bacterial community responses to warming

The bacterial community responded differently to warming at spatial (root, rhizosphere and bulk soil) and temporal (tillering, jointing and ripening stages) scales. For example, warming considerably affected the bacterial community composition in roots compared with the rhizosphere and bulk soil at an early wheat development stage (Figs. 3 and 4A). Proteobacteria (sum of the Alphaproteobacteria, Betaproteobacteria, Deltaproteobacteria and Gammaproteobacteria) was the most abundant phylum across the control and warming treatments and was relatively higher in the roots (47–55%) than in the rhizosphere (38–43%) and bulk soils (24–29%). In the root, rhizosphere and bulk soil compartments, warming increased the abundance of Alphaproteobacteria at the tillering stage and decreased Betaproteobacteria at the tillering and jointing stages. Cyanobacteria were substantially more abundant in the roots (6–26%) than in the rhizosphere (0.2–0.6%) and bulk soils (0.3–1.9%), while Bacteroidetes was more abundant in the rhizosphere (13–19%) than in the roots (3.3–11%) and bulk soils(3.3–4.1%). The relative abundances of Acidobacteria and Planctomyces increased considerably with distance from the roots. The relative abundance in response to warming was calculated (Fig. 4B). The relative abundances of Actinobacteria increased in response to warming and responded more significantly in the rhizosphere and bulk soil. In the warming treatments, Acidobacteria and Alphaproteobacteria increased at the tillering stage, while Gammaproteobacteria increased in the root and rhizosphere at the tillering and jointing stages (Fig. 4B).

In the root compartment, the relative abundance of the order Rickettsiales was considerably higher than that in the rhizosphere and bulk soil compartments and decreased

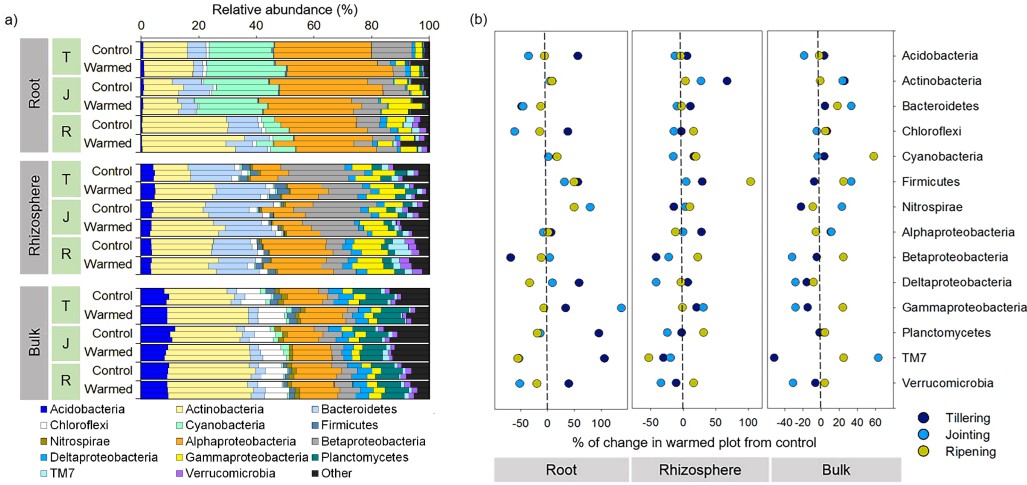

**Figure 4** Distribution of root, rhizosphere and bulk soil bacterial community compositions from the control and warmed treatments (A) and percent (%) of change in the warmed plot from the control at different plant growth stages (B). Percent change from the control; negative and positive values indicate a decrease and increase in the relative abundance of phyla in response to warming, respectively. T, tillering; J, jointing; R, ripening.

markedly from the tillering to ripening stages (Fig. S4). An increasing trend was observed for Rhizobiales from tillering to ripening in root and rhizosphere samples, while the abundance did not change significantly under warming. In the roots and rhizosphere at the tillering, jointing and ripening stages, the relative abundances of Actinomycetales, Bacillales and Pseudomonadales increased in response to warming, while the relative abundances of Sphingobacteriales and Burkholderiales decreased at the tillering and jointing stages in response to warming. Moreover, the abundance of Actinomycetales was higher in roots, while Sphingobacteriales and Burkholderiales were more abundant in the rhizosphere. In the roots and rhizosphere, the relative abundance of Rhizobiales increased with wheat development and was 2-3-fold higher at the ripening stage than at the tillering and jointing stages. The abundance of Rickettsiales was significantly higher in the roots (12–28%) than in the rhizosphere (0.2–0.4%) and bulk soil (0.06–0.31%).

Warming considerably influenced the bacterial genera at all wheat growth stages, and we analyzed the top 22 dominant bacterial genera (average relative abundance greater than 0.2%) (Fig. 5). In the roots and rhizosphere, the relative abundances of *Pseudomonas, Promicromonospora*, *Saccharothrix, Bacillus* and *Arthrobacter* increased after warming at the tillering and jointing stages. Moreover, the abundance of *Pseudomonas* was dramatically higher in the root and rhizosphere soil than in the bulk soil, and the abundances of *Devosia* and *Streptomyces* decreased with distance from the roots.

LEfSe analysis was performed to further identify differential species between the warming and control treatments in both root and rhizosphere bacterial communities. The differential species that met the linear discriminant analysis (LDA) significance threshold greater than 2.0 are shown in Fig. 6. The results confirmed that in the roots and rhizosphere, the phylum Actinobacteria, the order actinomycetes and the genera *Saccharothrix, Arthrobacter*,

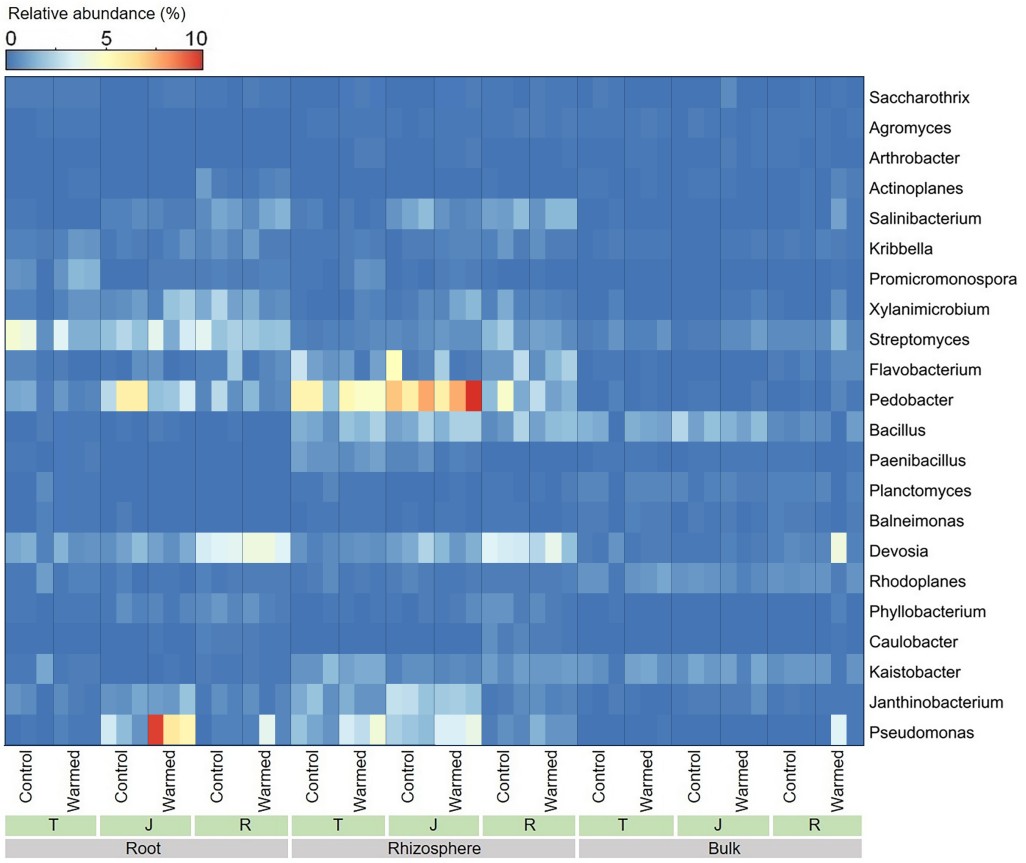

**Figure 5** **Heatmap of the dominant bacterial genera in root, rhizosphere and bulk soil samples.** T, tillering; J, jointing; R, ripening.

*Promicromonospora*, *Glycomyces* and *Cellulosimicrobium* increased significantly under warming conditions, and the relative abundance of the order Pseudomonadales and genus *Pseudomonas* also increased after warming.

## Functional prediction of bacterial communities

Differences in the function of the bacterial communities in the warming and control treatments of different root zone compartments and growth stages were assessed using PICRUSt2 with the Kyoto Encyclopedia of Genes and Genomics (KEGG) database. Six types of biological metabolic categories at KEGG level 1 were obtained (Fig. S5) with metabolism as the primary category (46.7%–50.0%). The functional profiles of energy metabolism and carbohydrate metabolism at KEGG level 3 were then predicted and plotted as a heatmap for comparison (Fig. S6). We found that for the carbohydrate metabolism, warming elevated the functional categories including galactose metabolism, ascorbate and aldarate metabolism, pentose and glucuronate interconversions, pentose phosphate pathway, and pyruvate metabolism in the rhizosphere.

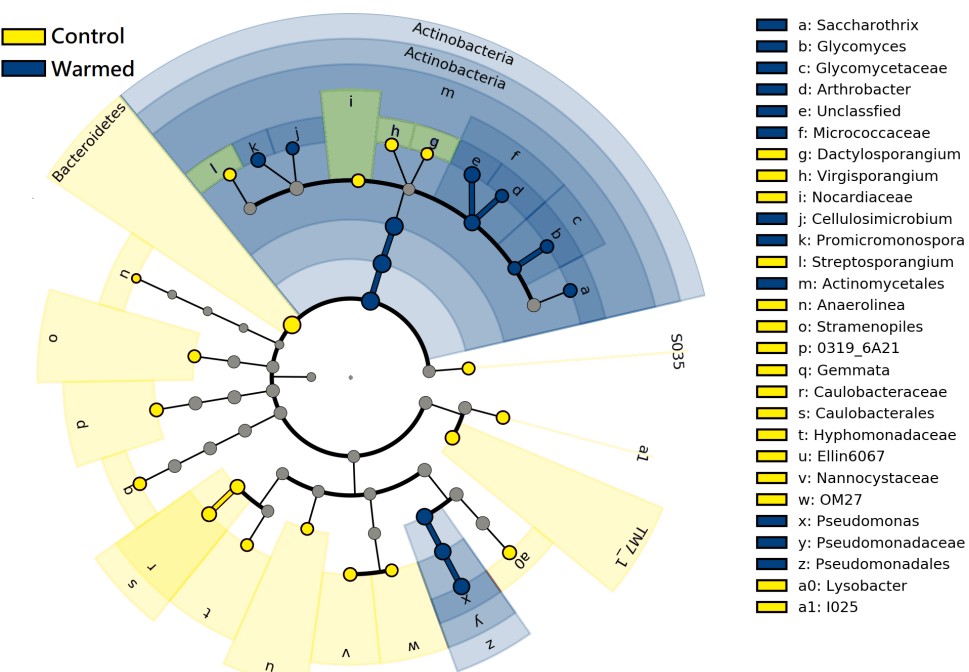

**Figure 6 Linear discriminant analysis Effect size (LEfSe) cladogram of comparing microbial communities between control and warmed treatments ($P < 0.05$, LDA > 2.0).** The circles from inner to outer stand for phylum, class, order, family, and genus. Green circles stand for taxa that were significantly abundant in the warmed treatments, red circles stand for taxa that were significantly abundant in the control treatments, and yellow circles indicate species with no significant change between the warmed and control treatments.

## DISCUSSION

Stronger responses of the microbial community structure (Fig. 3) and diversity and richness (Shannon, Simpson and Chao1 indexes) to warming were observed in the root and rhizosphere samples than in the bulk soil. This phenomenon could be due to the selective effect of roots on microbes (*Bulgarelli et al., 2013*; *Marilley & Aragno, 1999*; *Weisskopf et al., 2005*) and rapid turnover of the root exudate carbon through bacterial breakdown (*Weisskopf et al., 2008*). In roots, bacterial diversity (Shannon and Simpson) indexes were significantly decreased by warming at the ripening stage, while at previous stages, the effect was not significant (Fig. 2). The bacterial community structure was also clearly separated according to the plant growth stage in the root and rhizosphere samples (Fig. 3). Consistently strong effects of vegetative stage on the root zone microbial communities of wheat were also discovered in previous studies (*Chen et al., 2019*; *Donn et al., 2015*). Thus, soil warming, wheat developmental stage and root proximity were the major drivers structuring the microbial community compositions in the current study.

The relative abundance of microbial phyla was significantly affected by warming (Fig. 4); in particular, the phyla Actinobacteria and Firmicutes increased considerably under warming across all the investigated growth stages in the roots and rhizosphere (Fig. 4B).

*Hayden et al. (2012)* reported that a 2 °C increase in grassland soil temperature resulted in a significant increase in the abundances of Actinobacteria and Firmicutes. Actinobacteria are considered one of the most important decomposers in soils (*Subramaniam et al., 2016*; *Větrovský, Steffen & Baldrian, 2014*) and are less sensitive to heat stress due to their spore-forming ability compared with other phyla (*Hayden et al., 2012*). Moreover, in the roots, the abundance of Actinobacteria and its order Actinomycetales increased considerably with wheat development, and higher dominance was observed at the later wheat growth stage, which may be associated with the ability of this group to survive on a variety of complex substrates (*Watt et al., 2006*). Likewise, previous studies reported a higher abundance of Actinobacteria in older plant roots (*Donn et al., 2015*; *Thirup, Johnsen & Winding, 2001*; *Watt et al., 2006*). Similarly, Firmicutes, mainly represented by the genus *Bacillus,* increased in response to warming. *Bacillus* generally play an important role in the mineralization of plant-derived material and humus in soil (*Singh et al., 2019*), and a number of strains in this species have demonstrated strong heat tolerance and plant growth-promoting activities (*Bokhari et al., 2019*; *Ghosh et al., 2009*). The consistent increase in these taxa in response to warming in the root and rhizosphere samples suggested that these bacteria could be good candidates for making wheat more resilient to a climate change scenario.

Microbial diversity and community composition in the root zone play an important role in nutrient cycling and are sensitive to alterations in substrate availability (*Bai et al., 2017*; *Maestre et al., 2015*). With respect to the growth kinetics and substrate affinity for metabolism, microbes can be classified as copiotrophs and oligotrophs (*Ho, Di Lonardo & Bodelier, 2017*). Copiotrophic bacteria are characterized by a faster growth rate but lower substrate affinity, while oligotrophic bacteria have slower specific growth but stronger substrate affinity (*Chen et al., 2016*). Copiotrophic microorganisms respond rapidly to nutrient availability (*Li et al., 2021*) and preferentially use easily available soil organic carbon (*Chen et al., 2016*). Root exudates, composed of monosaccharides, glucose, organic acids, etc., are well-known sources of soil labile organic carbon (*Panchal et al., 2022*). A strong influence of root exudates on the microbial community structure has been demonstrated in prior studies, particularly the enrichment of copiotrophic bacterial populations (*Adamczyk, Rüthi & Frey, 2021*; *Zhou et al., 2019*). A similar phenomenon was also observed in the current study, the relative abundance of the genera Pseudomonas and Bacillus and the order Actinomycetales (including the genera Promicromonospora, Arthrobacter and Saccharothrix) increased in the warming plots (Fig. 5, Fig. S4), and all these taxa have been reported to be copiotrophic (*Cleveland et al., 2007*; *Goldfarb et al., 2011*; *Li et al., 2022*).

In the current study, warming elevated the DOC in the rhizosphere soil (Fig. 1). The input of organic carbon through rhizodeposition can alter the decomposition of soil organic carbon (SOC) through the rhizosphere priming effect (*Wang et al., 2016*). The increase in organic carbon may lead to the immobilization of soil nitrogen (*Cao et al., 2020*) and affect crop yield. Although PICRUSt2 analysis, a function prediction method that heavily depends on accurate gene annotations (*Langille, 2018*) indicated that warming enhanced several carbon metabolism processes, further study is needed to illustrate how warming effects microbial carbon metabolism in the investigated soils through functional assays.

## CONCLUSION

Overall, long-term experimental warming improved the availability of organic carbon in the rhizosphere and enhanced associated microbial activity. Importantly, warming exerted a stronger influence on the bacterial community structure in the root and rhizosphere compared to the bulk soil, and this phenomenon was observed across different growth stages. Microbial taxa in the phyla Actinobacteria and Firmicutes were found to persist in the warming treatments and were identified as candidates for making wheat more resilient to climate warming. This study provides new insights into the effects of climate warming on the recruitment and functioning of the microbial community in the root vicinity by altering root zone carbon availability in agricultural ecosystems.

## ACKNOWLEDGEMENTS

The authors thank the staff at the experimental station for managing the fields and Mr. Yang and Dr. Jiazhen Li for their assistance in sampling.

### Funding

This work was supported by the Strategic Priority Research Program of Chinese Academy of Sciences (XDB40020204), the National Key Research and Development Program of China (2021YFF1000400), and the National Natural Science Foundation of China (U22A60009). The funders had no role in study design, data collection and analysis, decision to publish, or preparation of the manuscript.

### Grant Disclosures

The following grant information was disclosed by the authors:
Strategic Priority Research Program of Chinese Academy of Sciences: XDB40020204.
The National Key Research and Development Program of China: 2021YFF1000400.
The National Natural Science Foundation of China: U22A60009.

### Competing Interests

The authors declare there are no competing interests.

### Author Contributions

- Jing Wang conceived and designed the experiments, performed the experiments, analyzed the data, prepared figures and/or tables, authored or reviewed drafts of the article, and approved the final draft.
- Shuaimin Chen performed the experiments, analyzed the data, authored or reviewed drafts of the article, and approved the final draft.
- Ruibo Sun performed the experiments, authored or reviewed drafts of the article, and approved the final draft.
- Binbin Liu conceived and designed the experiments, authored or reviewed drafts of the article, and approved the final draft.

- Tatoba Waghmode conceived and designed the experiments, performed the experiments, analyzed the data, prepared figures and/or tables, authored or reviewed drafts of the article, and approved the final draft.
- Chunsheng Hu conceived and designed the experiments, authored or reviewed drafts of the article, and approved the final draft.

## Data Availability

The raw data are available in the Supplemental Files and the sequence data is available at European Nucleotide Archive: PRJEB37653.

## Supplemental Information

Supplemental information for this article can be found online at http://dx.doi.org/10.7717/peerj.15428#supplemental-information.

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
