# Peer review of "Spatial and temporal dynamics of the bacterial community under experimental warming in field-grown wheat"

_PeerJ, doi:10.7717/peerj.15428_

## Round 0.1 · original submission · Major Revisions

The manuscript is interesting but needs to make serious revisions and address the questions raised by the reviewers. The English of the manuscript should be improved and the manuscript should be better organized and explained according to the suggestions of reviewers.

Reviewer 1 has requested that you cite specific references. You may add them if you believe they are especially relevant. However, I do not expect you to include these citations, and if you do not include them, this will not influence my decision.

Reviewer 1 ·

Basic reporting

General comments:

This manuscript focuses on investigating the effect of climate change on bacterial community composition and diversity using a field experiment.

However, the manuscript should be better organized in structure and further revised before it is published. Based on the current manuscript quality, this manuscript should be accepted with major revisions.

Experimental design

The research is original, and the research questions are well defined. The experiment scale and design are appropriate and sufficient to address the research questions.

Validity of the findings

General comments:
This manuscript focuses on investigating the effect of climate change on bacterial community composition and diversity using a field experiment.
The research is original, and the research questions are well defined. The experiment scale and design are appropriate and sufficient to address the research questions.
However, the manuscript should be better organized in structure and further revised before it is published. Based on the current manuscript quality, this manuscript should be accepted with major revisions.

Specific Comments:
Abstract:
The authors only need to cover the most important findings in their abstract. Consider writing it concisely and meeting the Peer J requirement (The number of words in the abstract).
Line 22: “May” is an ambiguous word. There are a lot of publications that support the evidence. Consider using “can”.
Line 22 “plant microbiomes, and Line 25 “plant microbiota”: In this manuscript, plant microbiomes and plant microbiomes seem to be the same meanings. If so, to make readers better understand this manuscript, I suggest that authors should use one term – “plant microbiomes” throughout this manuscript (please check the other sections such as Introduction, Materials and Methods, etc.)
Line 32 – 36: The authors mentioned the abundance of some bacterial taxa changed when temperature increased. The authors have done statistical analyses. I recommend authors to include p-values.
For example, “Microbial community composition, particularly the phyla XXX and XXX, shifted considerably (P<0.05) ….”
Line 39: “root-zone”. The term “root zone’ is used in this manuscript. Does “root zone” mean “rhizosphere”? If so, I suggest using a widely accepted term (i.e., rhizosphere) in this manuscript. This will make readers easily understand. If “root zone” and “rhizosphere” is not exchangeable, authors should clarify this in the manuscript.

Introduction:
Generally, the goal of the study is clarified. There is insufficient background information about this research. More background information is needed in the current study.
Line 43: “May” is an ambiguous word. There are a lot of publications that support the evidence. Consider using “can” or deleting “may”.
Line 50 - 51: “A thorough understanding of the mechanisms of ……”. This sentence is not clear and needs to be rewritten.
Line 75 – 76: I recommend adding another paragraph here. This manuscript is concentrated on the root and soil microbiomes related to the wheat species -- Triticum aestivum L. More information related to T. aestivum is needed. For example, authors should include the geographic distribution of this wheat species. Authors mentioned tillering, jointing and ripening stages of this wheat species. It is unclear about the growing season of this wheat. Previous studies related to the microbiomes of T. aestivum should also be included. The authors mentioned this wheat is a local winter wheat in Materials and Methods, but it is not clear the wheat distribution in China. If this is an important crop in China, the authors should mention the broad impact of this research.
Line 76: “We utilized a 10-year warming….”. It is not clear what this 10-year period spans. Consider revising to this – “We utilized a 10-year (from 20XX to 20XX) warming…”

Materials and Methods:
Generally, detailed information is needed.
Line 91-96: I suggest authors include a map of this field experiment showing the arrangement of each plot.
Line 114: “for DNA extracton”. Consider using “before DNA extraction”.
Line 116: “Soil variables and dehydrogenase activity”. Consider revising to “The measurements of soil variables and dehydrogenase activities”
Line 124: “Dehydrogenase enzyme activity is considered a marker of microbial activity”. Consider using “The dehydrogenase enzyme activity is considered a marker of microbial activities”.
Line 141: “341F:785R” is not the correct way to write a primer pair. Please include the whole primer sequence (5’ to 3’ direction).
Line 154 - 162: “Sequence processing and analysis….”.
In general, this paragraph related to QIIME analyses was not written clearly. I highly suggest checking the “Materials and Methods” sections in the following papers, and citing the following papers (These papers also used the software QIIME to do the analyses)
Jiang, X., et al. (2022). "Limits to the three domains of life: lessons from community assembly along an Antarctic salinity gradient." Extremophiles 26(1): 1-14.
Jiang, X., and Takacs-Vesbach, C.D. (2017) Microbial community analysis of pH 4 thermal springs in Yellowstone National Park. Extremophiles 21: 135-152.
Van Horn, D.J., Wolf, C.R., Colman, D.R., Jiang, X., Kohler, T.J., McKnight, D.M. et al. (2016) Patterns of bacterial biodiversity in the glacial meltwater streams of the McMurdo Dry Valleys, Antarctica. FEMS Microbiol Ecol 92.
Line 154: QIIME version is missing.
Line 159: It is unclear the OTU taxonomic assignment method, database name, and version. Please refer to the papers mentioned above.
Line 159-160: “subsampled (rarefied), and the alpha diversity was calculated based on the rarefied OTU tables”. Usually, both alpha diversity and beta diversity (PCA) require subsampling. If the authors did both, they should clarify here. It is not clear how many reads were subsampled from each sample.
Line 164: “An independent test (Student’s t-test)”. Consider revising to “A Student's t-test was used to…..”
Line 164-165: “R v4.0.3 (Team, 2014)”. If authors used other R packages to plot the graphs, the names of packages should be reported too. I suggest including the R scripts as part of supplemental materials.

Results:
In general, authors need to better organize each part in the Results section and make it easier to read.
I suggest reporting the results in the following order.
The first part – “Soil characteristics and microbial activity” includes current Line 172 – 177
The second part – “High-throughput sequencing reads and QC report” includes current Line 179- 183.
The third part – Please report “Bacterial diversity and richness” before “Bacterial community responses to warming”. This will include current Line 224 – 245.
The fourth part – Please move “Bacterial community responses to warming” to the end. This will include current Line 178 – 223.
Specific comments:
Line 226 – 227: “bacterial diversity (Shannon and Simpson) and richness (Chao1) indexes were determined ……”. Authors should include detailed results about these indices. For example, the authors should mention how these indices change among different groups. Are they increased or decreased? Are these changes significant? Please include the P values here.
Line 237-238: “The PCA showed greater separation for both the phylum and genus communities”. The authors reported the PCA analyses at both phylum and genus levels, but they did not mention these in the Materials and Methods. Please revise Line 167-170 and mention the taxonomic levels and the purpose of using two different taxonomic levels.

Discussion:
General comments:
The current discussion is focused on what they have found. However, the authors should link their experiment results to the “bigger picture” of global warming and ecological impacts.
Some recommendations:
First, authors can link the shift of bacterial communities (resulting from global warming) with wheat production. Does this shift lead to an increase or decrease in wheat production?
Second, the authors discussed copiotrophs and oligotrophs. Also, the authors measured soil dissolved organic carbon. How would the shift between copiotrophs and oligotrophs affect soil carbon loss? Based on this field experiment data, can authors discuss something about global warming and soil carbon loss? What roles do copiotrophic and oligotrophic microbes play in soil carbon metabolisms? Authors can refer to the previous similar experiment and published results.
Yergeau, E., et al. (2012). "Shifts in soil microorganisms in response to warming are consistent across a range of Antarctic environments." ISME J 6(3): 692-702.

Additional comments

None

Reviewer 2 ·

Basic reporting

The manuscript is well written with very clear and professional English. The state of the art of the knowledge in the field has been well explained. The hypothesis is relevant and the experiments are logically designed. The data and figures are well designed and explained.

Experimental design

The research question is well defined and relevant. The methods are well explained and answer the questions sufficient enough to generalize the findings and conclusions.

Validity of the findings

The data have been analyzed and conclusions are drawn logically. The manuscript is prepared with all statistics applied to satisfactory levels.

Additional comments

Over all the manuscript is prepared nicely with a lot of effort and the questions posed are valid. However, the discussion portion may be further refined. Especially, there are a lot of speculations and generalizations. The authors should be careful in that respect and avoid unnecessary exaggerations.

Reviewer 3 ·

Basic reporting

Elevated temperatures by climate change could affect agricultural crops in many ways. While it was reported that microbiota plays an important role in the environment and actively interacts with crops to help it against potential stress, little is known about how climate change could affect the behavior of crop microbiota that could subsequently impact crop growth and yield. In this work, Waghmode et al. performed field experiment in which they studied how elevated temperatures could affect soil composition and microbiota during different stages of crop growth. The manuscript is well-structured and well-written with all results clearly demonstrated and I only have a few related questions or comments here.

Experimental design

No comment.

Validity of the findings

Line-115: Please spell out DOC on first use in the manuscript: “dissolved organic carbon (DOC)”

Figure-1 to 3 and Discussion: it is exciting that the authors found that DOC is overall high during ripening stage and also the microbiome profile & diversity is more different in ripening stage, indicating there are potential crop-associated factors. Although the authors discussed how the potential reason for difference among root, rhizosphere and bulk soil, more discussion is needed to better address the difference in ripening stage (DOC and microbiome) and potential related mechanism.

Figure-2a and 3: it is great that the authors demonstrated generally consistent microbiome profiles between warmed group and control group with some interesting differences, suggesting their measurement is robust. My question here is: if this is averaged across replicates? If so, I would also suggest showing the biological replicates for these two figures to highlight the reproducibility of the result (maybe included in supplementary figure for Figure-2a and main figure for Figure-3)

Figure-3: I noticed that the sum of rows in figure-3 seems not to be 100% as the authors are showing the dominant genera only. Please include the criteria for defining dominant genera in figure legend and also include the dominant genera in bulk soil for comparison.

Additional comments

Moreover, I would suggest two quick additional analysis that greatly help show off the data generated by the authors:

1. using microbiome profile shown in Figure-2, the authors could perform LEfSe analysis (https://huttenhower.sph.harvard.edu/lefse/) to statistically estimate the enriched taxonomy under warm conditions for different crop growth stage and this analysis would provide more statistical power since the phylogenic structure are also considered.

2. Since the authors were performing 16S rRNA sequencing, it is difficult to get more gene-level information for pathway analysis. However, there is some existing tools (such as PICRUSt2) that the authors can use to infer how specific microbial pathway was altered under warmed condition which could provide mechanistic insight into the impact of these changes.

---

## Round 0.2 · accepted · Accept

The original Academic Editor is no longer available so I have taken over handling this submission.

I read carefully both the concerned reviewer points as well as the authors' corrections carefully. After consideration, I considered that the actual manuscript has all the suggestions of the reviewers and is now ready to be published.